# Defects and nanostrain gradients control phase transition mechanisms in single crystal high-voltage lithium spinel

Isaac Martens[1,5], Nikita Vostrov[1,5], Marta Mirolo [1], Steven J. Leake [1], Edoardo Zatterin[1], Xiaobo Zhu [2], Lianzhou Wang [3], Jakub Drnec [1], Marie-Ingrid Richard [1,4] ✉ & Tobias U. Schulli[1] ✉

Lithiation dynamics and phase transition mechanisms in most battery cathode materials remain poorly understood, because of the challenge in differentiating inter- and intra-particle heterogeneity. In this work, the structural evolution inside $Li_{1-x}Mn_{1.5}Ni_{0.5}O_4$ single crystals during electrochemical delithiation is directly resolved with operando X-ray nanodiffraction microscopy. Metastable domains of solid-solution intermediates do not appear associated with the reaction front between the lithiated and delithiated phases, as predicted by current phase transition theory. Instead, unusually persistent strain gradients inside the single crystals suggest that the shape and size of solid solution domains are instead templated by lattice defects, which guide the entire delithiation process. Morphology, strain distributions, and tilt boundaries reveal that the $(Ni^{2+}/Ni^{3+})$ and $(Ni^{3+}/Ni^{4+})$ phase transitions proceed through different mechanisms, offering solutions for reducing structural degradation in high voltage spinel active materials towards commercially useful durability. Dynamic lattice domain reorientation during cycling are found to be the cause for formation of permanent tilt boundaries with their angular deviation increasing during continuous cycling.

High-voltage spinels are promising cathode materials for next-generation Li-ion batteries due to their high energy density, and rapid (de)lithiation kinetics. $LiMn_{1.5}Ni_{0.5}O_4$ (LMNO) spinel is among the most interesting of these high-voltage, Co-free oxides, in part due to the ease of growing large, micrometer sized single crystals[1,2]. Single crystal cathode materials such as LMNO are expected to solve certain drawbacks of more conventional cathodes consisting of nanosized particles including gas release and structural degradation of secondary particles during longer cycling[3]. Mechanical strength of single crystals cathodes and lower number of grain boundaries allows to prevent cracking of the material, subsequent pulverization and capacity loss[4–7]. Lower surface area also leads to suppression of gas release[8,9]. Due to

such properties it is anticipated that single crystal cathode will significantly improve the cycling life of batteries allowing for several thousands of charge-discharge cycles without significant capacity loss[10].

Larger crystallites intrinsically create longer Li diffusion lengths, resulting in complex concentration gradients inside the active material[11]. While the phase transitions of LMNO have been investigated using X-ray and neutron powder diffraction[12–15], Li transport at the single particle level of most intercalation materials remains poorly understood[16–19]. Controlling Li concentration gradients is critical for maximizing the durability and charge rate capability of all single crystal cathode technologies. Unfortunately, imaging the traffic of Li

[1]ESRF - The European Synchrotron, 71 Avenue des Martyrs, 38000 Grenoble, France. [2]College of Materials Science and Engineering, Changsha University of Science and Technology, Changsha 410114, China. [3]Nanomaterials Centre, School of Chemical Engineering, and Australian Institute of Bioengineering and Nanotechnology, University of Queensland, Brisbane, QLD 4072, Australia. [4]Université Grenoble Alpes, CEA Grenoble, IRIG, MEM, NRX, 17 rue des Martyrs, 38000 Grenoble, France. [5]These authors contributed equally: Isaac Martens, Nikita Vostrov. ✉e-mail: mrichard@esrf.fr; schulli@esrf.fr

intercalation at the nanoscale is extremely difficult, despite a growing number of innovative strategies[20]. A holistic understanding of the phase transitions in these complex materials will need to incorporate heterogeneity at the particle, electrode, and device levels. A deeper mechanistic insight into charging mechanisms of high voltage oxides is necessary toward controlling their phase transitions, and enhancing their durability toward commercial use.

Komatsu et al. showed that the fully lithiated ($Li_1$, $Li_1Ni_{0.5}Mn_{1.5}O_4$), half-lithiated ($Li_{0.5}$, $Li_{0.5}Ni_{0.5}Mn_{1.5}O_4$) and delithiated ($Li_0$, $Ni_{0.5}Mn_{1.5}O_4$) phases are not completely segregated, and that nonstoichiometric, crystalline solid solution intermediates are formed during charge and discharge[12]. The quantity of these metastable solid solution intermediates scale with charging rate, but disappear after relaxation, and were therefore proposed to form at the grain-boundary-like interface between the three stoichiometric phases. The formation mechanism and location of the solid solution phases could not be determined, since intra- and inter-particle heterogeneity are challenging to distinguish using bulk powder diffraction. Investigations of phase transitions at the single crystallite level have been reported using X-ray spectroscopy[18,19,21], but not under potential control, to the best of our knowledge.

Scanning X-ray diffraction microscopy (SXDM) is an imaging technique where a nanofocused X-ray beam is rastered across a sample, measuring diffraction from each position on the crystal (Fig. 1a). Maps are collected over a narrow range of angles covering the diffraction peak of a microcrystal, creating a 3D reciprocal space map for each pixel of the 2D image (examples of slices through reciprocal space are presented in Supplementary Fig. 1). These maps are extremely sensitive to local lattice strain and crystallographic rotation/tilt of the measured planes (Fig. 1c). This relatively new technique[22] has been recently used to image the strain and defect structure of battery cathode microcrystals, but again never in situ[23–26]. Bragg coherent diffraction imaging (BCDI) has emerged as an alternative method for mapping strain inside LMNO cathode particles in situ[16,27–30]. However, significant practical limitations on maximum crystallite size, crystal quality, beam damage, reconstruction accuracy, measurement speed, and the difficulty of data analysis have limited the utility of this competing technique[31,32].

In this work, high-resolution SXDM is used under operando conditions on battery materials[27], to map the delithiation of a single LMNO crystal. We show how LMNO single crystals exhibit complex solid solution Li gradients at the nanoscale, even when charged at a low specific current of 10.25 mA g$^{-1}$ (C/15 or one full charge in 15 h). In opposition to the reaction-front model[12], solid solution domains are observed to encompass entire crystallites several microns in size. Strain gradients inside a single crystal appear nearly static during the first voltage plateau ($Ni^{2+}/Ni^{3+}$), indicating that the local state-of-charge (SOC) heterogeneities in the solid solution likely arise from intrinsic lattice defects. During the second voltage plateau ($Ni^{3+}/Ni^{4+}$) the gradients are highly dynamic, indicating fundamentally different mechanisms for the two phase transitions, and the influence of crystal defects on phase transformation. Understanding the origins of this strain/state-of-charge heterogeneity and local lithium distribution inside crystals will allow better control over their phase transitions, which is necessary to design cathode active materials with higher cycling stability and rate capability.

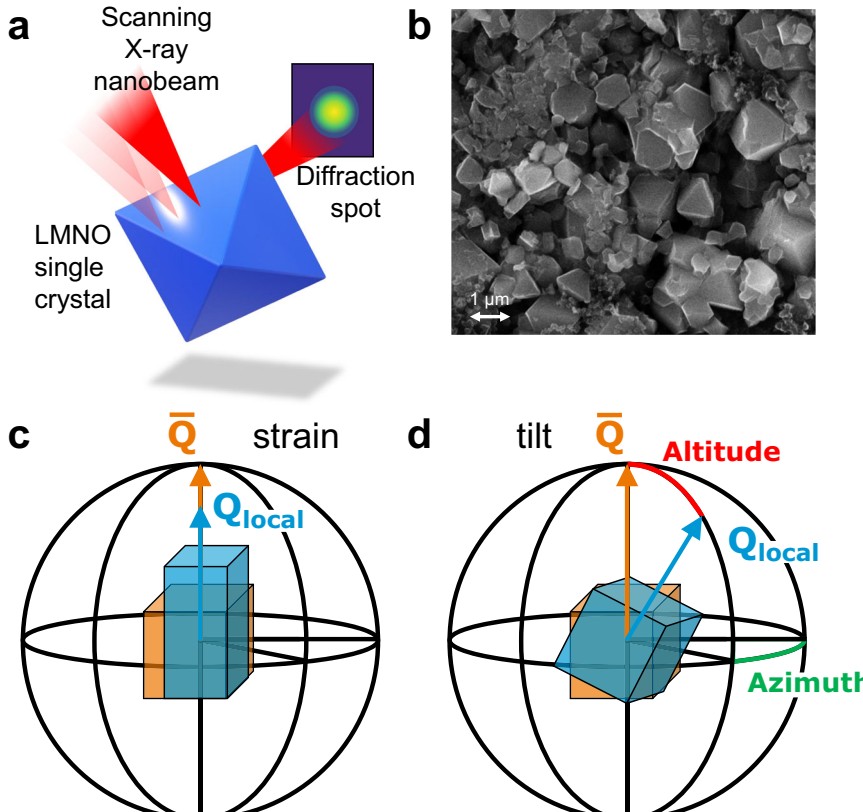

**Fig. 1 | Scanning X-ray diffraction microscopy (SXDM) overview.** Schematic of SXDM technique (**a**). The nanofocused beam is rastered across the sample, imaging the diffraction from different positions on the crystal. Electron microscopy image of the (111)-faceted, octahedra LMNO ($LiMn_{1.5}Ni_{0.5}O_4$) crystals (**b**). SXDM is sensitive to changes of local scattering vector **Q**$_{local}$ (blue) relative to the scattering vector averaged across the measured particle $\overline{\mathbf{Q}}$ (orange). Strain in the lattice causes deflection in the angle of the diffracted beam and changes the magnitude of the scattering vectors (**c**). Local tilt of the crystal lattice rotates the position of the diffracted beam leading to rotation of the scattering vectors that can be described by altitude (red) and azimuth (green) angles (**d**).

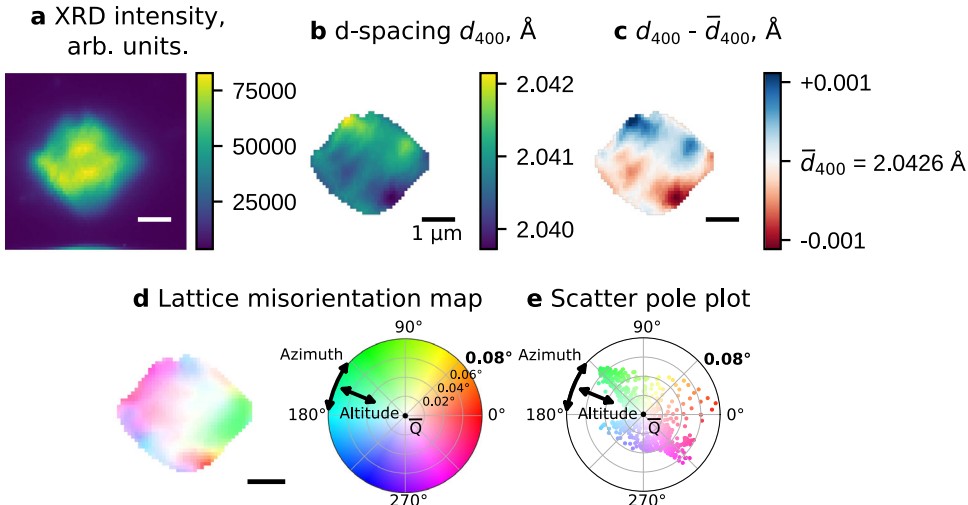

**Fig. 2 | Nanodiffraction imaging of LMNO single crystal before the start of the charge.** For each pixel of the map several parameters are calculated: the X-ray diffraction (XRD) intensity (**a**), the $d_{400}$ d-spacing between the (400) planes (**b**), variation of the d-spacing from its average value across the whole particle ($\bar{d}_{400}$) (**c**), azimuth and altitude of the tilt vector relative to the average orientation of the crystal ($\bar{Q}$) (**d**). The angular tilts for each pixel can also be plotted as a pole figure (**e**).

## Results and discussion

SXDM imaging of LMNO was performed in a commercial, reflection geometry Li‖LMNO operando cell with carbonate-based electrolyte and a kapton window for X-ray transparency. Additional description and information on the preparation of the cell is presented in Methods section. A (111)-faceted LMNO single crystal (Fig. 1b) was located, and 2D maps of the (400) reflection were acquired with 80 nm resolution (Fig. 2).

The strain and tilt for each pixel of the map is extracted from the position of the reflection. The diffracted intensity map (Fig. 2a) displays the integrated signal observed. Strain inside the particle can be displayed either as the variation in d-spacing (Fig. 2b), or more conveniently as the local deviation of the d-spacing from the mean value of the whole crystal (Fig. 2c). Mosaic domains inside the single crystal create tilt boundary defects, which can be thought of as grain boundaries with very small misorientations (≤0.2°). The direction and magnitude of these tilts can be expressed with azimuth and altitude angles of a spherical coordinate system, the main axis of which points along the average Q-vector of the particle ($\bar{Q}$), and mapped directly onto the crystal (Fig. 2d). The hue represents the direction of the tilt (azimuth), while the saturation of the color conveys the magnitude of the tilt vs. the average orientation (altitude). The tilts from each pixel can also be plotted in the form of a pole figure, which is convenient for tracking the distribution of tilts over time (Fig. 2e). Detailed description of SXDM data analysis is presented in "Methods" section.

In addition to excellent sensitivity toward crystalline ordering, nanodiffraction has two advantages over soft X-ray[33] and electron beam[34] spectromicroscopy for in situ imaging of battery materials. While spectroscopic imaging follows phase transitions through changes in Ni oxidation state, it cannot differentiate solid-solution and ordered phases[18,19]. Second, diffraction experiments can be performed at arbitrarily high X-ray energies. This allows for (1) penetration through sample environments up to several centimeters thick[35] providing compatibility with industrially relevant electrodes, and (2) no need for spectroscopic deconvolution or background subtraction, leading to extreme precision toward Li intercalation and (3) potentially lower beam-induced damage[36]. Given the well-known difficulty, cost, and low success rates of in situ soft X-ray and electron microscopy experiments, the convenience and ease of use offered by SXDM is extremely attractive.

It is worth noting that in the SXDM setup configuration, the detector distance was maximized to ~1.4 meters. This deliberate adjustment aimed to enhance the strain sensitivity of the measurement. However, this setup choice comes with a trade-off, as it limits the field of view in the reciprocal space and restricts our ability to investigate the presence of the amorphous phase that may form on the phase boundaries. Indirectly, we can examine the existence of the amorphous phase by analyzing the measured intensity around the primary Bragg reflection from the crystalline portion of the particle (see Supplementary Fig. 10). Interestingly, two peaks in this plot coincide with phase transitions and suggest the occurrence of amorphization in a portion of the particle. However, accurately quantifying the fraction of the amorphous phase is challenging due to the limited field of view of the detector during the experiment.

Several LMNO particles of various sizes were measured ex situ at different reflections (**400**, **111** and **511**) before charging the cell (Fig. 2 and Supplementary Figs. 2–5, 11). The normalized strain dispersion (often termed microstrain) for different particles ranges from 600 to 1100%% while the tilt inhomogeneities vary from 0.05 to 0.13°. We did not observe any significant dependencies of strain and tilting from the size of the particles. These microstrains, along with the characteristic, gradient-like pattern of the strain maps suggest an inhomogeneous Li distribution during the synthesis process. In contrast, the observed tilt boundaries appear to accommodate these slight differences in lattice parameter at different points of the crystal (Fig. 2d, e and Supplementary Figs. 2–5d, e). Operando SXDM maps were collected during galvanostatic charging, starting from the lithiated state, and following the delithiation over the potential plateaus between 4.7 V to 4.75 V associated with the $Ni^{2+}/Ni^{3+}$ and $Ni^{3+}/Ni^{4+}$ redox couples, up to a potential of 4.85 V.

Figure 3 illustrates the structural sensitivity advantage of SXDM strain imaging over spectroscopy. In the top row, images of the crystal during the $Li_1/Li_{0.5}$ phase transition are shown with a color bar spanning the full range of d-spacings observed throughout the first phase transition, corresponding to a total change in strain of about 1.5% (2.04 Å to 2.01 Å). The internal microstructure of the crystal in the top row is invisible, and appears to possess a homogeneous lithiation state which shifts over time. No reaction front or phase boundaries are visible, and the solid solution appears to consume the whole particle, with none of the original fully lithiated phase remaining. The bottom row displays the same images, but with a color map encompassing only

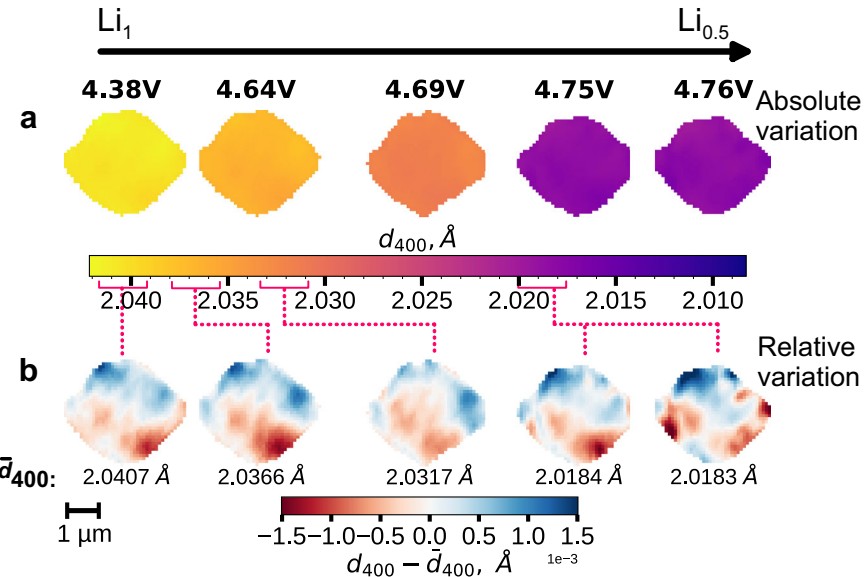

**Fig. 3 | Operando nanodiffraction images of an LMNO single crystal at different potentials during the $Li_1 \longrightarrow Li_{0.5}$ phase transition.** The color bar for the top row of images uses the complete $d_{400}$ d-spacing range observed during charging (**a**). In the bottom row, the same snapshots are colored showing only variation within the crystal relative to average d-spacing ($\bar{d}_{400}$), allowing the instantaneous strain and lithiation heterogeneity throughout the crystal to be visualized (**b**).

the relative variation of the d-spacing inside each frame. Subtle differences in the strain of the crystal in each image are immediately visible. The reciprocal space resolution of the images is at least $2.5 \times 10^{-4}$ Å, which (assuming linear strain dependence[12]) corresponds to resolvable differences in lithium stoichiometry of 0.012%. These very small shifts are impossible to detect using spectroscopy at the Ni edge, where the lithiation state accuracy is typically ±3%[37], even for high quality ex situ measurements. The extreme sensitivity of high-resolution nanodiffraction imaging toward strain and defects allows the lithiation pathways inside crystals to be imaged in situ, with unprecedented levels of detail.

### $Li_1 \longrightarrow Li_{0.5}$ phase transition

The strain evolution inside the LMNO crystal throughout the first voltage plateau is shown in Fig. 4a, images 4–14. The average d-spacing is given above each image. The gradual evolution from 2.0409 Å at 4.1 V to 2.0332 Å near 4.7 V is in good agreement with previous operando powder diffraction studies[12,38]. The observed strain corresponds to the convolution of the spatially heterogeneous lithiation of the LMNO, together with lattice distortion induced by any crystallographic defects[29]. Histograms from each frame (Fig. 4c) allow SXDM data to be directly compared with in situ powder diffraction data. The histogram peak widths (also known as microstrain) obtained during charging (presented in Supplementary Fig. 6) quantify the SOC heterogeneity inside the particle. The SOC heterogeneity is at a minimum in the relaxed, initial state before charging the cell. The detector was positioned to observe only one of the LMNO phases ($Li_1$ or $Li_{0.5}$) at a given time. White pixels on the edges of the particle correspond to regions where the phase transition is complete and the imaged phase is no longer detected.

The internal microstructure and strain distribution is nearly constant during the initial part of the charging curve (Fig. 4a, images 1–9), despite delithiation shifting the average d-spacing of the particle. In the latter stages of the first voltage plateau (Fig. 4a, images 10–14), phase segregation is triggered, and the strain pattern inside the crystal evolves rapidly. Multiple nucleation points of the $Li_{0.5}$ phase are detected, and the $Li_1$ phase is consumed in a three-dimensional fashion. The nucleation points for the $Li_1 \longrightarrow Li_{0.5}$ phase transition appear correlated with the most delithiated regions on the surface of the crystal (the deepest red pixels). These regions also exhibit increased standard deviation values which are observed from the beginning of charge (Supplementary Fig. 9, images 1 and 9) indicating clusters of persistent defects inside the particle.

The nucleation points are not clearly associated with corners or edges of the crystal. Regardless, it is clear that the $Li_1 \longrightarrow Li_{0.5}$ phase transition follows a "phase-field" mechanism, growing from multiple points on the surface. This opposes the popular "core-shell model" used to describe delithiation, where a Li-poor shell slowly consumes a shrinking Li-rich core[37]. During the last image of the first transition (Fig. 4a, image 14) after the majority of the crystal has converted to the $Li_{0.5}$ phase, there are still small regions which have not oxidized. These regions appear to be selectively localized at the edges and corners of the crystal. Edges and corners cause additional energy barriers through domain-wall-pinning and coherency strain during phase transitions[39,40], which explains why these regions of the crystal are the last to transform. The nature and shape of the propagating reaction front is important toward understanding phase transitions, and is controlled by both the Li diffusion rate through the lattice, and the exchange current density at the crystal surface. Inherent crystal defects in LMNO appear to play a major role in the reaction front propagation inside single crystals. During this study the distance between the sample and the detector was maximized to improve the reciprocal space resolution, however this resulted in a limited field of view in the reciprocal space. The same experiment was repeated later on another particle (not shown here) to investigate a larger volume of the reciprocal space (see ref. 41). While other unique phenomena related to lattice rotation were detected, we observed similar phase transformation mechanisms for different particles.

Kuppan et al. observed that the (100) truncated edges of octahedral crystals served as nucleation points for the phase transition[18]. In Fig. 4a, it is difficult to precisely identify the number and locations of the nucleation points, given the "depth-summed", 2D projected image of the 3D crystal. 3D nanoimaging is necessary to link phase transitions with specific surface sites or facets because of the limited spatial resolution and surface sensitivity. Several ex situ or ensemble-averaged studies (especially those produced by chemical delithiation) have suggested that the phase transitions occur in a unusual, triphasic mechanism, with fully lithiated, partially lithiated, and

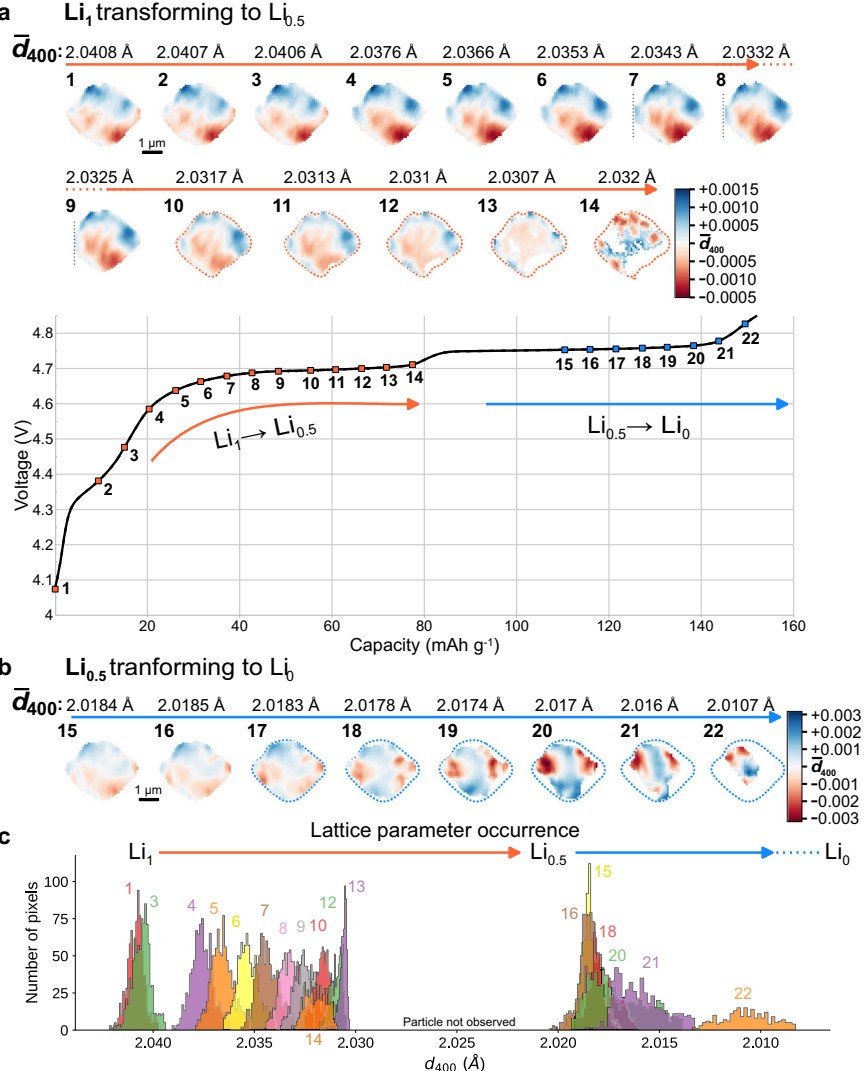

**Fig. 4 | Charging curve for the Li||LMNO cell at 10.25 mA g⁻¹ (C/15) and strain maps of the measured particle collected during operando diffraction microscopy.** Each d-spacing map corresponds to a point along the charging curve when it was collected and an average d-spacing of the crystal ($\overline{d}_{400}$) at this point of charging (presented above the maps). For LMNO, the first voltage plateau at 4.7 V (marked by orange arrows) corresponds to the Ni²⁺/³⁺ redox couple and Li₁ ⟶ Li₀.₅ phase transition (**a**). The second voltage plateau at 4.75 V (blue arrows) corresponds to the Ni³⁺/⁴⁺ couple and Li₀.₅ ⟶ Li₀ phase transition (**b**). The dotted lines represent the outline of the particle taken from image 10. Evolution of the $d_{400}$ histogram of the particle during charging (**c**). Several histograms are omitted for clarity (full figure is presented in Supplementary Fig. 16).

completely delithiated phases simultaneously coexisting within the same LMNO crystallite[12,18]. However, this model is not supported by our data, or other in situ reports on single particles[16]. The much slower, electrochemical oxidation used here understandably exhibits a more homogeneous microstructure, and better represents the reaction fronts found inside operational batteries[42].

## Li₀.₅ ⟶ Li₀ phase transition

Since only one of the Bragg reflections was measured at each scan, we can rely on the evolution of intensity (Supplementary Fig. 10) to confirm the complete transformation of the diffracted volume into the following Li₀.₅ phase.

The second phase transition and voltage plateau occur at 4.75 V (Fig. 4b, images 15–22). These images show a clear evolution in the strain gradients inside the crystal, unlike those from the first voltage plateau. The regions with locally enriched and depleted Li concentration fluctuate, even as the average d-spacing remains unchanged at 2.0184 Å (images 15–17). These dynamics reflect the equilibration

and redistribution of Li between phases of similar energy inside the crystal. In the final four images of this sequence, the crystal progressively shrinks (images 19–22). This reflects the transformation of the outer regions of the particle to the Li₀ phase, which was not imaged. Unlike the Li₁ ⟶ Li₀.₅ transition, the Li₀ phase grows over the surface of the crystal, forming a core-shell morphology as the reaction progresses. In the last frame recorded at 4.82 V, a small central Li₀.₅ core remains inside a Li₀ shell (image 22). During the last stages of the second transition, the range in d-spacing values across different regions of the particle reaches nearly 0.006 Å, which is double the variation observed during Li₁ ⟶ Li₀.₅ transition. The different morphologies of the intermediates (phase-field vs. core-shell) and the larger strain gradients created during the second phase transition support the idea that these two reactions follow different mechanisms at the nanoscale. Large internal strain gradients in a crystal are intrinsically linked to the formation of defects and structural collapse[29], which may be responsible for the poor stability of LMNO at high potentials.

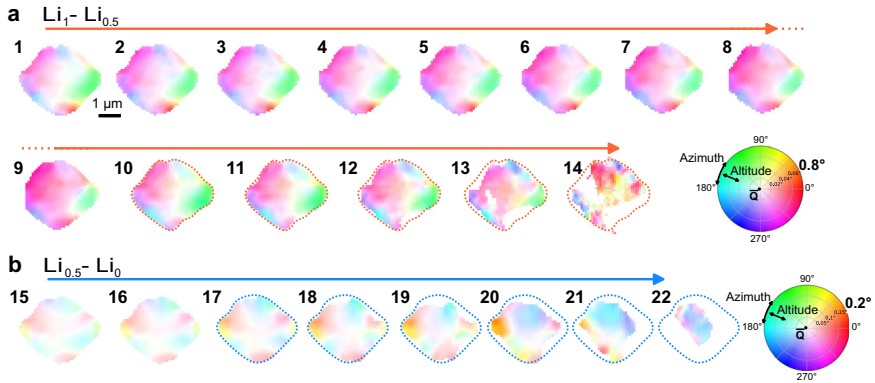

**Fig. 5 | Nanodiffraction imaging misorientation maps of the LMNO collected during charging.** The top two rows were collected during the first voltage plateau ($Li_1 \longrightarrow Li_{0.5}$ transition) (**a**), while the bottom row was collected during the second voltage plateau ($Li_{0.5} \longrightarrow Li_0$ transition) (**b**). The hue and saturation on the colorwheel correspond to the azimuth and the altitude of the scattering vector, in the polar coordinate system shown in Fig. 1c. The numbering matches the data shown in Fig. 4. Note the tilt magnitude for part (**b**) is larger than in (**a**), to accommodate the larger tilts seen during the second voltage plateau.

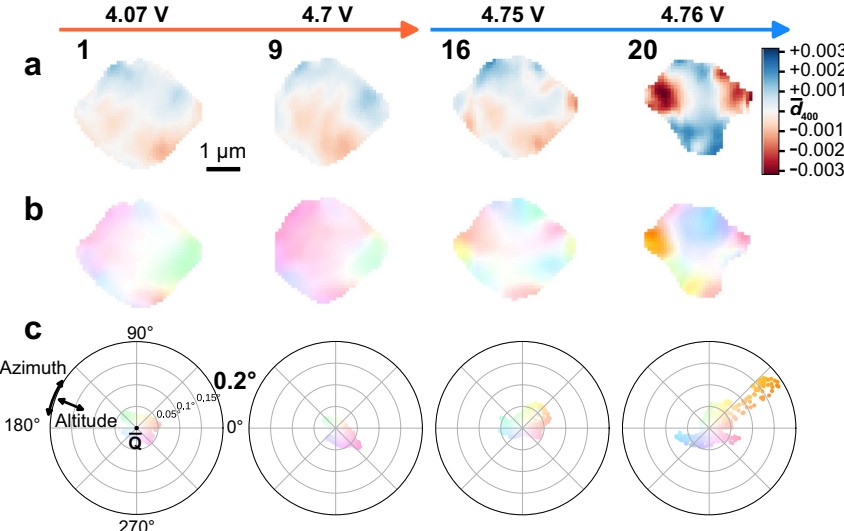

**Fig. 6 | Nanodiffraction imaging maps of the beginning and end of the $Li_1 \rightarrow Li_{0.5}$ (images 1 and 9 respectively) and $Li_{0.5} \rightarrow Li_0$ (images 16 and 20) phase transitions.** They include the maps of ($d_{400}$) d-spacing (**a**), misorientation maps of the LMNO particle (**b**) and scatter polar plots (**c**) where each point represents the azimuth and altitude deviation of local Q-vector at each pixel of the map. Colorbars for the maps and the polar plot range are set to the full range observed during both of the phase transition. Note that image 9 is cut on the left due to slight misalignment of the particle relative to the scan area.

## Defects and mosaicity

Understanding the interplay between lithiation gradients from defects inside the particle is challenging from strain maps alone. Luckily, SXDM is not only sensitive to strain, but also crystallographic orientation. Defects and mosaicity inside the crystal lattice can lead to tilt boundaries, previously reported as a common type of defect in $LiCoO_2$[23–25] and LMNO[43]. The evolution of the misorientation maps during both phase transitions is shown in Fig. 5 with two colorwheels, each corresponding to one of the phase transitions (a version with unified colorwheel is presented in Supplementary Fig. 7). Differently colored regions correspond to tilted, mosaic domains inside the LMNO single crystal. The corresponding pole figure plots for all images can be found in Supplementary Fig. 8. To aid comparison, the strain, tilt, and pole figure plots for several images are presented together on the same scale in Fig. 6.

Charging the cell alters the relative orientation of these mosaic nanodomains. In the initial stages of charging, slightly misoriented domains inside the crystal can be observed, with angular deviations up to 0.08° (image 1). The boundaries between tilt domains are surrounded by white space (zero average tilt), implying that a gradual transition in tilt occurs between them. During the charge, the top left segment of the particle (magenta region) starts to reorient and the white space vanishes, forming a more pronounced, sharper interface between itself and the green, bottom right region of the particle (images 2–9). Similar processes also occur between the blue regions at the top and bottom of the crystal, near the end of the first phase transition.

At the beginning of the second phase transition, the color pattern of the crystal is completely different than in the initial state, indicating reorganization of the mosaic domains (Fig. 5, a vs. b). The regions where the misorientation vector deviates the most from the average vector (saturated colors) are now located on the edges of the particle while the large central region of the particle is homogeneously oriented along the average Q vector (white color). The tilted domains are also smaller, more numerous, and the magnitude of their relative tilt distortions are now much larger, up to 0.25° (Fig. 6). This correlates with the higher d-spacing variation during the second phase transition (Fig. 4b and Supplementary Fig. 6). The relationship between the

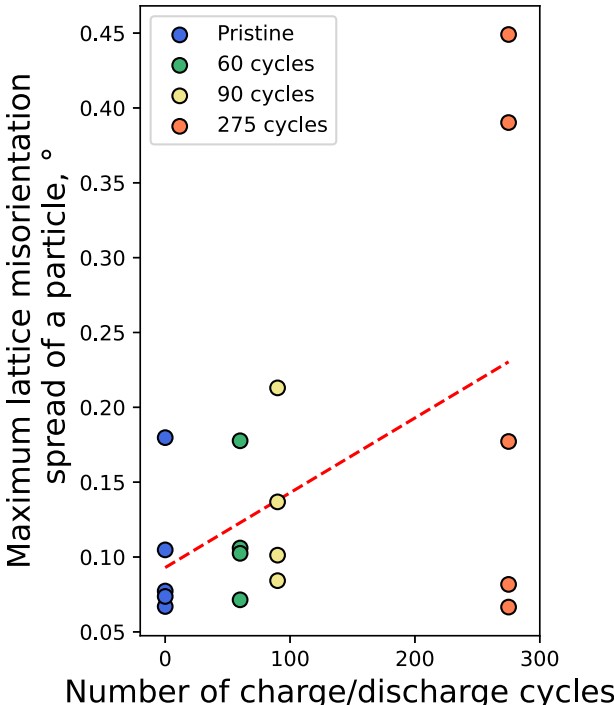

**Fig. 7 | Evolution of lattice misorientation in LMNO particles upon cycling.** Maximum angular lattice misorientation between two points of the map for particles of the pristine and cycled LMNO samples (60, 90 and 275 cycles).

boundaries are persistent throughout the phase transitions. The mosaic tilt domains inside a crystal dynamically rotate and reorient relative to one another during charging, probably to relieve strain induced by volumetric contraction of the delithiated crystal. These observations are consistent with previous in situ X-ray analysis which detected crystal rotation and defect persistence during cycling[16]. The influence of mosaicity on the lithium diffusion inside the crystals is difficult to predict from first principles. While defects are normally thought to interrupt ion intercalation pathways, several kinds of defects have been shown to locally enhance Li transport in LMNO, forming a network of conductive channels inside crystals[44,45]. The lack of strain evolution in the solid solution region indicates (at least at 10.25 mA g$^{-1}$ (C/15)), that the tilt boundaries with angles less then around 0.1° do not impede Li transport.

Several maps of the particles of varying sizes (1–3 from samples subjected to different number of charge-discharge cycles (0, 60, 90 and 275) were obtained to assess the long-term impact of sub-particle fine misorientation features (Supplementary Figs. 11–14). Figure 7 demonstrates a noticeable trend of increasing maximum angular deviation. Notably, larger tilt inhomogeneities, reaching up to 0.6° between the most misoriented lattice domains (compared to <0.1° in the control pristine sample as shown in Supplementary Fig. 11) were found in certain particles from the cycled sample (Supplementary Fig. 14). This confirms that the observed dynamic reorientation of the lattice seen operando leads to a continuous development of permanent low-angle tilt boundaries during consequent cycling. Furthermore, the growth of the misorientations is highly heterogeneous over the population of measured particles, and only some particles seem to be affected over time. Persistent defects present in the initial population of particles guide the long-term structural evolution of the material as was observed in our study[41]. Cycling likely causes defective particles to grow more defective over time, while defect-free particles remain defect-free. The increased internal resistivity at such boundaries[46–50] is likely to play a crucial role in the observed capacity decrease (Supplementary Fig. 15) during continuous cycling, while even longer cycling will likely lead to formation of cracks inside the crystal. Similar lattice misorientation domains were also observed in pristine single crystal NMC622 cathode particles[51].

Further work is needed to understand whether internal tilt boundaries are desirable from a stability perspective, since their reorganization can both accommodate strain, but also serve as nucleation points for microcracks or locally enhance dissolution. Nevertheless, we anticipate control over this new dimension of nanostructure and intracrystal domain dynamics will be critical for understanding the influence of doping, cation-ordering, and coatings toward the stability and rate-capability of high voltage spinels in general.

lattice strain and tilt domains is complex, and most of the visible domains do not directly map onto each other. However, during the first voltage plateau several Li-poor regions on the bottom of the particle, are colocalized with a sharp tilt discontinuities (Fig. 6 and Supplementary Fig. 9, images 1 and 9). These areas correlate with the locations of sharp increase in the standard deviation (Supplementary Fig. 9, images 1 and 9) and, at the later stages they also coincide with the origin points of the phase transformation (Fig. 4, images 11–14). Such tilt boundaries may serve as the nucleation points for phase transitions, as previously suggested[28,43].

The larger spatial strain heterogeneity across the whole particle during the second phase transition is accompanied by a substantial increase in local nanoscale strain heterogeneity, evidenced by higher peak widths at certain positions on the crystal and delocalization of such regions (standard deviation maps on Supplementary Fig. 9, image 16). Their placement is highly correlated with the aforementioned tilt domains, that are formed at the surface of the particle (Fig. 5, images 15–20). This characteristic distribution of tilt domains over the surface of the particle and nanoscale strain variation could facilitate Li diffusion, and lead to the preference for a more homogeneous delithiation.

Despite the variation between strain and mosaicity maps, the orange-yellow tilt domain, that can be seen at images 17–21 of Fig. 5, exactly coincides with the Li-poor area of the particle (Fig. 6, image 20). Moreover, the angular deviation in orientation of this domain gradually increases during charging (Supplementary Fig. 8, images 17–21) together with the decrease of the local d-spacing in the same region (Fig. 4, images 17–21). This further indicates that domain misorientation and defects facilitate interfacial Li transport. This indicates that tilt boundaries emerging between misoriented domains could hinder delithiation due to higher resistance and therefore form sharp local Li-ion concentration gradients.

Several mechanistic conclusions can be drawn from these strain and tilt maps. The LMNO does not "recrystallize", from the solid solution during phase segregation, and many defects including tilt

## Methods

### Material synthesis
Disordered, single crystal LMNO was prepared by a polymer assisted sol-gel method: 0.02 mol of nickel(II) acetate tetrahydrate, 0.06 mol of manganese(II) acetate tetrahydrate and 0.042 mol of lithium acetate dihydrate were dissolved in 150 ml of distilled water, followed by adding 0.12 mol of acrylic acid. The solution was stirred under 90 °C until dry gel was formed. Then the dry gel was heated at 400 °C for 3 h and grounded into fine powder using a mortar. Afterwards, the powder was calcinated at 750 °C for 8 h to get LNMO. All chemicals were purchased from Sigma-Aldrich unless otherwise stated.

### Electron microscopy
Electron microscopy images were recorded with a Hitachi TM3000 benchtop scanning electron microscope (SEM) in backscatter electron mode with an incident beam of 15 keV.

## Electrochemical cycling

The active material was made into electrodes using a doctor blade onto Al foil (99.6%, 15 μm) thick, Guangdong Canrd Ltd), with an ink composition of 85 wt% LMNO, 10% conductive carbon, and 5% poly-vinylidene fluoride (Solef PVDF 5130/1001, Solvay) binder with N-methyl pyrrolidone (NMP) serving as the solvent. The electrodes were dried at 110 °C in vacuum for 12 h and then compressed at a pressure of 10 MPa. The loading mass of the electrodes is $4 \pm 0.5$ mg cm$^{-2}$. A 0.3 mm thick Li foil (Alfa Aesar, 99.9% purity) was used as the counter and reference electrode, while a 1 mm thick Whatman glass fiber separator (GF-A) was used to provide gentle mechanical compression of the electrode stack. Lithium hexa-fluorophosphate solution in ethylene carbonate, dimethyl carbonate and diethyl carbonate (1.0 M LiPF$_6$ in EC/DMC/DEC; 1/1/1 v/v/v, LP-57, Sigma-Aldrich) was used as an electrolyte. Long-term cycling was car-ried out using 2016 coin-cells with the same components as described above with electrolyte volume of 60 μl. OCV for these cells ranged from 3.98 to 4.07 V. Electrochemical testing protocol consisted of 2C charge (187.2 mA g$^{-1}$) of the cell until 4.8 V, 10 min voltage hold, 10 min relaxation, 2C discharge, until 3.5 V, 10 min voltage hold and 1 h relaxation. Cells were cycled for 60, 90 and 275 cycles. Cycling was carried out in the temperature range of 22–24 °C. The voltage range utilized in this experiments goes beyond the operating voltage of conventional electrolytes including the one used in this work. To determine the degradation mechanism of the electrochemical system coin-cell cycled to 275 cycles was reassembled with new electrolyte and Li anode, but with the same cathode used before. Since we didn't see an increase in capacity after reassembly with new electrolyte we can conclude that the electrolyte decomposition is not the only cause for degradation of elements of the studied system and structural changes to the cathode also plays an important role in the capa-city fade.

## XRD operando cell preparation

For operando cycling the electrochemical cell sample environment (Opto-Std) was purchased from EL-CELL, and used without modification with 150 μl of electrolyte. During the measurement the cell with OCV equal to 4.05 V was charged at 10.25 mA g$^{-1}$ (C/15) until 4.9 V. The commercial cell was heated in a vacuum oven at 80 °C and at less than 5 mbar overnight to remove water. The glass fiber separators and cathode were baked out in the same way, but at 120 °C. A kapton window of 230 μm thickness was used to provide X-ray transparency over the 10 mm diameter observable area. The cell stack was oriented in the conventional fashion, such that the active film of the cathode was facing the separator and Li anode, and not toward the kapton window. The cell was constructed inside a Ar-filled glovebox with O$_2$ and H$_2$O levels below 0.5 ppm. An SP-240 Biologic potentiostat was used to control the cell current and record the potential. The temperature of the cell was controlled at 21.0 °C during imaging. No cycling was performed before the X-ray experi-ment, but the cell was allowed to rest for several hours. The open circuit potential was monitored and became constant ~10 min after cell assembly.

## Synchrotron measurements

Synchrotron powder diffraction of the pristine electrode ex situ, as well as in situ diffraction over the first charge-discharge cycle of an Li‖LMNO cell, confirm the electrode performance and active material were consistent with previous literature reports. Detailed analysis of these results have been presented elsewhere[38]. Rietveld analysis of the uncycled LMNO scraped from an electrode yielded the same lattice parameter within error as the SXDM. The lattice parameter of the LMNO can be calculated by multiplying the d-spacing of the 400 reflection reported here by a factor of 4.

The images in this work were obtained from an individual single crystal under operando conditions. However, several other crystals in the cell were imaged at open circuit potential prior to cycling, and from the as-fabricated electrode sheet in ambient air. The data from all of these crystals appeared qualitatively similar (Supplementary Fig. 11).

SXDM maps were collected at beamline ID01 at the European Synchrotron Radiation Facility. The X-ray wavelength was 1.24 Å (10 keV, 19° angle of incidence), with a beam size of 80 nm, and a flux of 10$^9$ ph/s, focused using a Fresnel zone plate. A 512 × 512 pixel Maxipix detector at a distance of 1.6 m was used to collect the nanodiffraction data. Detailed descriptions of the microscope[52] and mapping technique[22] have been described previously. 2D images were collected at 30 angles with a step size of 0.016° to map the local d-spacing of the reflection across the crystal. This produces a 5-dimensional dataset with each pixel of the 2D image corresponding to a 3D rocking curve. One full 5D scan of a 6 × 6 μm$^2$ area took ~30 min.

X-ray damage is a serious concern for operando synchrotron experiments. Because strain microscopy requires accurate peak posi-tions, but not peak intensities, even noisy diffraction data from low-dose imaging with short exposure times produce excellent images. To minimize the influence of radiation damage, the X-ray beam exposure per pixel was kept to 10 ms or lower, re-imaged every 90 s. This dose is similar or less than previous in situ hard X-ray nanoprobe experiments where no damage was observed[28], and orders of magnitude less than transmission spectromicroscopy experiments with soft X-rays or full-field lenses. No flux dependent phenomena, such as beam-induced crystal rotation was detected during the experiment. Consensus on the influence of radiation damage and suitable dose limitations in various operando battery chemistries has yet to emerge[53].

## SXDM data analysis

Python scripts utilizing the XSOCS and SXDM libraries developed at ESRF (https://kmap.gitlab-pages.esrf.fr/xsocs/; https://gitlab.esrf.fr/id01-science/id01-sxdm-utils) were used to analyze the data. Conver-sion of the diffracted intensity to reciprocal space was done by "bin-ning" the diffracted intensity data in each voxel of an array with dimensions of 50 × 100 × 100 and applying 2 × 2 median filter. The 3D fitting of the center of mass of a Bragg peak was used to determine the direction and the length of local Q-space vectors. The images were masked to exclude pixels with less than 20% of the maximum diffrac-tion intensity.

An example of the reciprocal space maps generated by SXDM is shown in Supplementary Fig. 1. The top left image shows the integrated intensity of one particle measured during the charge, with a selected pixel marked with a red cross. The other three plots show projections of the 3D reciprocal space corresponding to this individual pixel, viewed through the XY, XZ and YZ directions. The red crosses in each of these maps reflect the fitted maximum of the diffraction peak, used to calculate the strain and tilt values. Supplementary Fig. 6 displays the evolution of the peak width (along the rocking curve direction) during charging. Supplementary Fig. 7 displays the misorientation maps identical to Fig. 5 in the main text, but on a unified color scale to aid comparison. Supplementary Fig. 8 shows the pole figure scatter plots for all the SXDM frames. Note the altitude range of the plots for images 1–14 is from 0° to 0.08° and from 0° to 0.2° for images 15–22.

## Data availability

The operando[54] and ex situ[55] SXDM data generated in this study are openly available through the ESRF Data Portal.

## Code availability

XSOCS and SXDM python libraries that were used to analyze the SXDM data are openly available on Gitlab (https://kmap.gitlab-pages.esrf.fr/xsocs/; https://gitlab.esrf.fr/id01-science/id01-sxdm-utils).

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

## Acknowledgements

The authors thank Hamid Djazouli and Irina Snigireva for expert technical support, and Claire Villevieille for the donation of the electrolyte. This work was supported by the European Research Council under the European Union's Horizon 2020 research and innovation programme, grant agreement numbers 814106 (TEESMAT) and 818823 (M.I.R., CARINE). X.Z. acknowledges support from the National Natural Science Foundation of China (52202210). L.W. appreciates the financial support from Australian Research Council through its Linkage Project (LP1701000392).

## Author contributions

N.V., I.M., M.M., S.J.L., E.Z., M.I.R., and T.U.S. conducted the X-ray experiments and analysis. X.Z. fabricated the active material. I.M., L.W., J.D., M.I.R. and T.U.S. conceived the experiment. All authors reviewed the manuscript.

## Competing interests

The authors declare no competing interests.
