## [Peer Review File · Nature Communications]

Defects and nanostrain gradients control phase transition mechanisms in single crystal lithium spinelREVIEWER COMMENTS

Reviewer #1 (Remarks to the Author):

The manuscript introduces high-resolution Scanning X-ray Diffraction Microscopy (SXDM) as novel technique to monitor and correlate in real time the evolution of local strain and Li-distribution in a single crystal material during battery operation. The authors could observe a change in delithiation mechanism upon different voltage plateaus for LMNO, which they put in context with literature and explain where their model deviates from other theories (triphasic mechanism and core-shell). The manuscript is well received, aims to explain the fading mechanism of LMNO with a high quality of data and hence is recommended for publication upon minor revision.

1) Could the authors please elaborate more on the limitations of such technique? Surely amorphous materials will fail to deliver diffraction peaks, whereas amorphization is also an important feature during charge/discharge. Especially when observing complex systems with grain boundaries, which would be of great importance for studies of all-solid-state batteries, I wonder about the limitations of SXDM. So is the technique only useful for intrinsic studies on single crystalline materials surrounded by liquids or what is the future perspective?

2) What is the advantage of large single crystals as cathode materials in battery applications? I cannot gather that from the manuscript, on the contrary when referring to line 21.

3) Can you clarify in line 113, line 137 and line 175 what is surface for you? If I understand correctly we are just observing direction (400), but how can you conclude on surface and near-surface/bulk effects?

4) Has there been any efforts made to compare different sizes of particles ranging from nano- to microscale? Could the authors please elaborate if they would expect any significant changes in the (de-)lithiation mechanisms proposed?

5) In the following I suggest some minor changes for better understanding and revisit for typos: Line 18 insert "promising cathode materials". Figure 1b, I would highlight the particle studied in the manuscript for clarity. Line 80 insert "different points of the crystal" and please reference to the corresponding image (Figure 2a, b, c, etc.) for better reading. Line 130 small formatting issue. Line 76 I cannot find the (511) reflection in Figures S2-S5 in the Supporting Information. Further Figure S2a states (004) plane in the description, other than indicated in the image itself.

Reviewer #2 (Remarks to the Author):

The authors conducted nano-beam XRD mapping in situ on a single LNMO particle during the charging stage in the first cycle. The results showed that the pristine particle had an initial intrinsic micro-domain structure that might induce delithiation heterogeneity in the stages from Li1 to Li0.5 and Li0.5 to Li0. The authors also found that the delithiation development in the stage Li1 to Li0.5 did not follow the commonly assumed core-shell model but was affected by the pre-existing micro-grain structure.

The reviewer finds the results interesting and providing certain insights into the delithiation behavior in the first charging cycle. The authors also demonstrated that low-dose nano-beam XRD might be a valuable tool for in situ battery characterization. However, the reviewer also feels that the current results are not sufficient to support some statements made by the authors. The current measurements only show the initial Li1 and intermediate Li0.5 states, and the Li0 state is not provided. Furthermore, from the reviewer's point of view, it is more important to show if any initial micro-grain structures in the pristine particles will be inherited in the following cycles and hinder long-term performance. As indicated by the bulk XRD measurement in ref [4] cited in the manuscript, after the discharge in the first cycle, Li1 is fully reinstalled and has a sharper and taller peak than the pristine. It suggests that micro-grain structures in the pristine particles are modified, and the crystallographic defects are reduced. It is also unclear if there is still (de)lithiation heterogeneity in a perfect pristine particle after the first cycle. Although it is probably safe to claim that (de)lithiation heterogeneity causes the long-term cyclability issue, which is a trivial statement and has been made by other groups as well, it is still unclear if this is an issue related to the initial structure or simply the intrinsic material properties.

Some minor concerns are shown below:

1. The particle shape changed during Li1 to Li0.5, even in the early frames. The width of the particle gradually decreased, and the corner on the left-hand side seemed to be truncated. What is the reason for that? Is that due to sample motion, or does that suggest some level of core-shell progress?
2. XRF mapping would be helpful in determining the sample shape and boundary, especially with the detectable particle domain changes along the cycling in the current measurement scheme. XRF mapping would be very helpful in determining if the particle is still in the same place. XRF mapping can be done simultaneously with XRD mapping.
3. The results based on a single particle are useful but lack statistical rigor. It would be helpful to show results of multiple particles.

The reviewer feels that the manuscript, in its current form, is not suitable for publication in NC. The reviewer suggests a major revision before it can be accepted in the future.

Reviewer #3 (Remarks to the Author):

This manuscript reported a novel operando SXDM technique to observe the phase transition of high voltage $\text{LiMn}_{1.5}\text{Ni}_{0.5}\text{O}_4$ single crystals during the delithiation process, which would be great helpful to understand the origins of the strain or state-of-charge heterogeneity and local lithium distribution inside crystals, to better control over their phase transitions, so that designing a cathode active material with higher cycling stability and rate capability. Therefore, I think this work is technique advanced and would be constructive to the investigation of battery materials, especially for single crystal cathodes. I would recommend this manuscript to this journal. However, there are still some comments that should be considered.

1. More details about the electrodes and test conditions should be mentioned, e.g. the thickness and areal loading of positive electrodes.
2. What happens inside the cathode between 4.7-4.75 V, the authors skipped this voltage part, but if compare the last image (14th) during Li_1 to $\text{Li}_{0.5}$ and the first image (15th) during $\text{Li}_{0.5}$ to Li_0 , a very different strain variation is observed. It would be helpful if more images could be offered.
3. This work mainly focuses on delithiation process, why did not show a full cycle including lithiation as well, which could better understand the influence of strain variation to particles upon cycling.
4. Actually, most battery materials belong to polycrystals, e.g. polycrystalline nickel-rich cathodes, there are many grains inside one particle, the anisotropic strains easy resulting in the generation of microcracks, how this in situ SXDM technique to detect and imaging the strains along boundaries for these materials.

Response to Reviewers:

Manuscript: NCOMMS-23-02491-T

Title: **Defects and nanostrain gradients control phase transition mechanisms in single crystal battery cathodes**

Reviewer #1:

The manuscript introduces high-resolution Scanning X-ray Diffraction Microscopy (SXDM) as novel technique to monitor and correlate in real time the evolution of local strain and Li-distribution in a single crystal material during battery operation. The authors could observe a change in delithiation mechanism upon different voltage plateaus for LMNO, which they put in context with literature and explain where their model deviates from other theories (triphase mechanism and core-shell). The manuscript is well received, aims to explain the fading mechanism of LMNO with a high quality of data and hence is recommended for publication upon minor revision.

We thank the Reviewer for his/her constructive comments.

- 1) *Could the authors please elaborate more on the limitations of such technique? Surely amorphous materials will fail to deliver diffraction peaks, whereas amorphization is also an important feature during charge/discharge. Especially when observing complex systems with grain boundaries, which would be of great importance for studies of all-solid-state batteries, I wonder about the limitations of SXDM. So is the technique only useful for intrinsic studies on single crystalline materials surrounded by liquids or what is the future perspective?***

Scanning X-ray diffraction microscopy (SXDM) measures the diffraction signal of a sample with a highly focused X-ray beam. In our manuscript, we focus on a very small angular window around a Bragg reflection with extreme resolution, which is appropriate for modelling the fine structure inside highly ordered materials such as battery microcrystals, or for example semiconductor samples. Less crystalline materials will produce broader reflections, but with identical total intensity. This diffuse scattering can be measured by pushing the detector closer to the sample, measuring a broader angular range, albeit with lower angular resolution. This configuration is routinely applied on several of our beamlines to investigate fully or partially amorphous samples including bone/shell tissues, nanocrystalline electrocatalysts (10.1021/acscenergylett.1c00718), or even solvation shells in liquids (10.1126/science.1261412).

Several types of disorder at the single particle level can be distinguished. If a single crystal domain splits into multiple crystalline grains, it is useful to consider the standard deviation of the peak position for all the voxels in the particle, such as in Figure S6 of the revised manuscript. Nanoscale disorder and amorphization (smaller than individual pixels in the map) can be evaluated through the peak width of a reflection, or by looking at the off-Bragg intensities surrounding the reflection. The new Figure S10 shows the evolution of the diffuse halo outside the central peak corresponding to the main crystalline particle. The figure exhibits two maxima, which both coincide with the phase transitions and indicate that part of the particle is becoming amorphous during the transition. However, it is challenging to calculate the precise fraction of the amorphous phase due to limited detector field of view and spatial

resolution of the beam. This analysis has been incorporated into the discussion in the revised version of the manuscript.

The SXDM technique is useful for *in situ* and *operando* structural studies in complex liquid, solid or gas environments. Hard X-rays allow far deeper penetration into materials and with much less beam damage than electron probes, with higher resolution than optical or neutron techniques. The main limitation of any high resolution, single particle, *in situ* technique is to what extent the measured particle captures a representative picture of the entire population in the sample. This is why this study used identical conditions to those we employed previously with *operando* powder diffraction to study the average crystal structure (reference 38, 10.1021/acsmaterialslett.2c00787).

In addition, we have now better explained these possibilities in the discussion, and measured several additional crystallites (*ex situ*) from the same electrode to understand the structural diversity present in these complex materials (Figs. S11-14).

2) *What is the advantage of large single crystals as cathode materials in battery applications? I cannot gather that from the manuscript, on the contrary when referring to line 21.*

We appreciate the reviewer pointing out this missing context. Cycling creates enormous repetitive strains inside active materials as Li ions enter and exit the lattice. In conventional polycrystalline materials, this anisotropic strain becomes concentrated at grain boundaries, and is well-understood to induce cracking and pulverization of the particles. Inside a device, this leads to progressively lower storage capacity and power performance through a number of mechanisms. Large single crystals without grain boundaries and defects largely eliminate these phenomena, potentially extending the practical useful lifetime of Li-ion batteries up to several hundred years (see: J.E. Harlow, *et al.*, 10.1149/2.0981913jes).

Additional text in the has been added to the introduction to explain the great interest in the emerging technology of single crystal cathodes (lines 20-26).

3) *Can you clarify in line 113, line 137 and line 175 what is surface for you? If I understand correctly we are just observing direction (400), but how can you conclude on surface and near-surface/bulk effects?*

In all these instances, we are referring to the near-surface regions observed around the perimeter of the 2D projection of the particle in the SXDM images. The use of this term has been clarified in the text, since the ~ 80 nm resolution of the images considers mostly the bulk by the standards of the surface science community. (line 147) True 1-2 nm surface sensitivity can indeed be measured for materials which are inactive in the bulk, as we have demonstrated in a recent study (10.1038/s41563-023-01528-x).

4) *Has there been any efforts made to compare different sizes of particles ranging from nano- to microscale? Could the authors please elaborate if they would expect any significant changes in the (de-)lithiation mechanisms proposed?*

In our experience, this comparison is complicated because crystal size and shape is often strongly correlated with the crystal quality, and both of these parameters can be directly linked to (de)lithiation mechanisms. It is extremely challenging to grow crystals with different sizes or shapes while controlling their defect content, especially since LMNO exhibits a continuum of cation ordered phases with nearly equivalent free energy.

From a mechanism perspective, the capability to map lithiation gradients in nanocrystals is limited by the spot size of the beam, in this case 80 nm. For typical polycrystalline battery materials (~50 nm LFP/LTO, 300nm NMC/NCA) this leads to very few pixels in the image. Ongoing improvements in microscope resolution are now routinely expanding capabilities down to 2-5 nm resolution (DOI: 10.1364/OE.25.025234), for strongly diffracting samples with heavy atoms. The operando experiments necessary to probe mechanism are limited by beam damage, which increases dramatically at high magnifications (proportional to resolution squared), comparable to in-situ TEM experiments.

In a new study (DOI: 10.26434/chemrxiv-2023-7t1td, also attached), we have examined ensembles of nano and micro particles with Multi Crystal X-ray Diffraction, building on the works of Singer *et al.* (DOI: 10.1021/nl502332b, reference. 16 in the revised version of the manuscript). This technique allows the powder diffraction of many individual particles to be independently tracked during cycling, including nanocrystals too small to map using SXDM.

In another new study on layered cathode materials ($\text{LiNi}_{0.8}\text{Mn}_{0.1}\text{Co}_{0.1}\text{O}_2$), we examined the dependence of particle size and shape using SXDM and nano-tomography methods (DOI: 10.26434/chemrxiv-2023-p621k, also attached). A large degree of lattice bending and crystal curvature was observed, attributed to layered oxygen vacancies, which increases with the linear size of the particles. This sort of relationship was not observed in LMNO, since its simpler cubic structure does not support layered defects and does not grow large anisotropic crystals.

For the present manuscript, we have added maps of 18 new particles with varying sizes (1-3 μm) inside uncycled and cycled cathodes (Figs. S11-14). Over this limited range of sizes visible in the sample, we do not observe any link between internal structure and particle size. Since no time was available for operando measurements on these particles, we cannot conclude on if there is any size dependence in phase transition mechanisms. However, a trend of increasing lattice misorientation was observed for cycled samples (Figure 7). A paragraph discussing these results was added to the main text (lines 95-96 and 216-226).

5) In the following I suggest some minor changes for better understanding and revisit for tips: Line 18 insert “promising cathode materials”. Figure 1b, I would highlight the particle studied in the manuscript for clarity. Line 80 insert “different points of the crystal” and please reference to the corresponding image (Figure 2a, b, c, etc.) for better reading. Line 130 small formatting issue. Line 76 I cannot find the (511) reflection in Figures S2-S5 in the Supporting Information. Further Figure S2a states (004) plane in the description, other than indicated in the image itself.

We thank the reviewer for their attention to detail in correcting these mistakes. We have modified the text to reflect the suggested changes.

Reviewer #2:

The authors conducted nano-beam XRD mapping in situ on a single LNMO particle during the charging stage in the first cycle. The results showed that the pristine particle had an initial intrinsic micro-domain structure that might induce delithiation heterogeneity in the stages from Li1 to Li0.5 and Li0.5 to Li0. The authors also found that the delithiation development in the stage Li1 to Li0.5 did not follow the commonly assumed core-shell model but was affected by the pre-existing micro-grain structure.

The reviewer finds the results interesting and providing certain insights into the delithiation behavior in the first charging cycle. The authors also demonstrated that low-dose nano-beam XRD might be a valuable tool for in situ battery characterization.

We thank the reviewer for his/her appreciative comments.

However, the reviewer also feels that the current results are not sufficient to support some statements made by the authors. The current measurements only show the initial Li1 and intermediate Li0.5 states, and the Li0 state is not provided.

That is correct, the microscope as configured during the experiment only had enough angular range to measure two of the three phases continuously. We selected to measure the Li₁ and Li_{0.5}, after which no more beamtime was available.

A new supplementary Figure S10 shows the evolution of the integrated intensity of the studied particle. At the beginning of charging, one can notice that the integrated intensity closely matches the intensity after the Li₁-Li_{0.5} phase transition. This indicates that the particle fully recovers into the Li_{0.5} phase excluding a possible amorphous phase, as discussed above.

The discussion regarding the evolution of Bragg reflection intensity has been added to the main text of the manuscript. (lines 153-154)

In a new manuscript, (DOI: 10.26434/chemrxiv-2023-7t1td, also attached), we repeated portions of this experiment in a configuration which allowed us to measure all three phases (Li₁, Li_{0.5}, Li₁). SXDM images confirm the results of this study, and highlight the dynamics of the low-angle tilt boundaries between domains inside a single crystal particle. The main objective of that work was to perform simultaneous SXDM and Multi Crystal X-ray Diffraction, which bridges single crystal measurements with powder diffraction on an ensemble of particles.

Furthermore, from the reviewer's point of view, it is more important to show if any initial micro-grain structures in the pristine particles will be inherited in the following cycles and hinder long-term performance.

We fully agree with the reviewer's reasoning here. Long-term cycling is an important issue for high voltage spinel materials such as LMNO, but aging phenomena are distinct from the focus on lithiation mechanism in this study, although the two are related. It is understandably very difficult to follow an individual particle over long term cycling, given the challenge of maintaining a perfectly aligned nanobeam over several days, and the very limited time available on SXDM instruments in synchrotron facilities.

To study the effects of long-term cycling on single-particle lattice structure, a separate *ex situ* experiment was conducted with cycled cathodes during which SXDM maps of several (different) particles were measured. Figures related to this experiment were added to the Supplementary Material (Figures S11-18) and the discussion was added to the main text (lines 216-226). We observe a progressive increase of the angular lattice misorientation in cycled samples, indicating the development of low-angle tilt boundaries during repetitive phase transitions in the cathode. Interestingly, the growth of these misorientations is highly heterogeneous, and only some particles seem to be affected over time. This is consistent with the reviewer's suggestion that persistent defects present in the initial population of particles guide the long-term structural evolution of the material. Cycling likely causes defective particles to grow more defective over time, while defect-free particles probably remain defect-free. An additional figure in the revised main text summarizes these results (Fig. 7).

As indicated by the bulk XRD measurement in ref [4] cited in the manuscript, after the discharge in the first cycle, Li1 is fully reinstalled and has a sharper and taller peak than the pristine. It suggests that micro-grain structures in the pristine particles are modified, and the crystallographic defects are reduced. It is also unclear if there is still (de)lithiation heterogeneity in a perfect pristine particle after the first cycle.

We thank the reviewer for catching this insightful detail.

The main advantage of diffraction microscopy is that we can resolve intracrystal heterogeneity, which is impossible with powder diffraction. Broader peak shapes could certainly originate from microstrain associated with defects as suggested by the reviewer. However, an ensemble of particles with a heterogeneous state of charge/lithiation would produce an indistinguishable strain pattern. Diffraction microscopy is poorly suited for characterizing intercrystal heterogeneity, because it is impractical to image more than a handful of crystals. It is useful to remember that LMNO crystals are grown by rapidly quenching from high temperatures, to prevent the formation of the thermodynamically preferred ordered phase (with lower performance), and because Li continuously evaporates during the synthesis. Cooling rates introduce defects, while Li evaporation influences strain/stoichiometry in the pristine material. Subsequent cycling vs a Li anode can backfill any Li inventory lost during synthesis.

To conclusively resolve this intra vs interparticle issue, our follow-up study uses simultaneous *operando* SXDM with Multi Crystal X-ray Diffraction (DOI: 10.26434/chemrxiv-2023-7t1td, also attached). In the latter technique, several hundred crystals are illuminated, and their individual speckles in the powder pattern are tracked over time. We show that both intra and intercrystal heterogeneity have significant contributions during the first cycle, and that the defects in the pristine material are both persistent and dynamic.

The microstructure of the pristine crystals seems very complicated, and grows even more intricate during cycling. These multiple considerations are partly what prompted us to define the scope of this current manuscript, which we believe remains an important contribution as the first demonstration of *operando* electrochemical SXDM.

Although it is probably safe to claim that (de)lithiation heterogeneity causes the long-term cyclability issue, which is a trivial statement and has been made by other groups as well, it is still unclear if this is an issue related to the initial structure or simply the intrinsic material properties.

We agree with the reviewer that this claim is safely supported by our data. We also agree that more work is needed to determine to what extent the crystalline microstructure contributes towards degradation phenomena. This is likely a strong function of (dis)charge rate, surface coatings, maximum voltage, and many other device operating parameters, as we previously demonstrated in reference 2 of the main text (DOI: 10.1038/s41467-022-28963-9). To definitively answer these questions on long-term aging, it will be necessary to image several crystals, dismount and cycle the cell repeatedly, and then locate and re-image the identical crystals during a different beamtime. We have had recent success performing this process with isolated crystals immobilized on a wafer, but have not yet discovered how to do this with battery coin cells. Repeated cycling induces nanoscale mechanical flexure, the crystals are not isolated, and are embedded deep inside the device, all of which make relocating a crystal difficult.

Some minor concerns are shown below:

1. The particle shape changed during Li1 to Li0.5, even in the early frames. The width of the particle gradually decreased, and the corner on the left-hand side seemed to be truncated. What

is the reason for that? Is that due to sample motion, or does that suggest some level of core-shell progress?

The particle indeed drifted several pixels during the overnight measurement, which is responsible for truncating the left-hand corner by 1-2 pixels, visible in frame 7-9, and realigned beginning in frame 10 in Figures 4-6. A brief explanation has been added in the experimental section. Any other broader changes in the particle shape detected by the reviewer are real, and reflect a loss in local crystallinity. In this work, we used a reasonably conservative threshold of 20% of the maximum intensity to mask the particle in each frame. More aggressive thresholding (5 or 10%) can be used in an attempt to gain better surface sensitivity and shape information, at the cost of lower quality fitting and significant noise.

2. XRF mapping would be helpful in determining the sample shape and boundary, especially with the detectable particle domain changes along the cycling in the current measurement scheme. XRF mapping would be very helpful in determining if the particle is still in the same place. XRF mapping can be done simultaneously with XRD mapping.

We have attempted simultaneous XRF-XRD mapping in this system to study typical electrodes. However, it has become evident that SXDM has a major advantage over XRF since only crystals in Bragg orientation diffract, while even slightly rotated crystals are invisible. Conversely, the XRF detects all crystals, regardless of orientation. For a typical electrode film with a thickness of 10 μm , the beam travels about 12 μm through the layer ($10/\cos(35^\circ)$). Given the broad particle size distribution centered at 1-2 μm , the incident beam excites fluorescence in many overlapping particles, and sees only a contiguous film. In theory an electrode film with a monolayer of isolated crystals would address this, but would dramatically decrease the quality of the electrochemistry, and likely alter any degradation phenomena. For this reason, we prefer to use electrodes fabricated with conventional loadings, thicknesses, and ink compositions. Furthermore, XRF was not available during the *in situ* experiment because the synchrotron was operating in a bunched timing mode incompatible with energy resolved detectors. Detecting the particle shape with XRD works quite well because the background signal is very, very low, essentially zero counts, as seen in Figure 2a.

Despite these limitations we agree that nano-XRF could be a valuable tool not only for confirming particle location but also for studying the distribution of transition metals inside the particles. For LMNO only the Ni edge would be visible *in situ*, since the Mn K-edge signal will not escape the coin cell.

3. The results based on a single particle are useful but lack statistical rigor. It would be helpful to show results of multiple particles.

A constant (and valid) critique of single particle imaging is the question of how representative the selected crystal is. To investigate this as well as the influence of continued cycling on the particles, we imaged 18 additional particles *ex situ*, the results from which are presented in Figures S11-14. We have also repeated the same *operando* experiment on another particle as part of a study (DOI: 10.26434/chemrxiv-2023-7t1td, also attached) to map a much larger range of reciprocal space. We observed similar behavior in this second particle, and our conclusions towards the phase transition mechanisms in LMNO are not affected.

The reviewer feels that the manuscript, in its current form, is not suitable for publication in NC. The reviewer suggests a major revision before it can be accepted in the future.

Reviewer #3:

This manuscript reported a novel operando SXDM technique to observe the phase transition of high voltage $\text{LiMn}_{1.5}\text{Ni}_{0.5}\text{O}_4$ single crystals during the delithiation process, which would be great helpful to understand the origins of the strain or state-of-charge heterogeneity and local lithium distribution inside crystals, to better control over their phase transitions, so that designing a cathode active material with higher cycling stability and rate capability. Therefore, I think this work is technique advanced and would be constructive to the investigation of battery materials, especially for single crystal cathodes. I would recommend this manuscript to this journal.

We thank the reviewer for his/her appreciative comments.

However, there are still some comments that should be considered.

1. More details about the electrodes and test conditions should be mentioned, e.g. the thickness and areal loading of positive electrodes.

Values for estimated thickness and the measured areal loading have been added to the methods section.

2. What happens inside the cathode between 4.7-4.75 V, the authors skipped this voltage part, but if compare the last image (14th) during Li_1 to $\text{Li}_{0.5}$ and the first image (15th) during $\text{Li}_{0.5}$ to Li_0 , a very different strain variation is observed. It would be helpful if more images could be offered.

Indeed, there is a gap in the data between images 14 and 15, where the microscope's detector and sample rocking curve angles were repositioned to measure the $\text{Li}_{0.5}$ phase. In this study, the microscope only possessed a field a view sufficient to observe a single phase at one time. We have added a statement better acknowledging this gap in the revised text of the manuscript.

In a new manuscript (DOI: 10.26434/chemrxiv-2023-7t1td, also attached), we present additional *in situ* SXDM imaging. We were able to carefully position the detector such that two phases were observed at once. Specifically, the Li_1 and $\text{Li}_{0.5}$ phases are measured simultaneously, followed by a short gap switching to the $\text{Li}_{0.5}$ - Li_0 transition. We hope to develop fully gapless imaging in the future, without compromising the extreme angular resolution necessary for visualizing the fine structure in these materials. New, larger detectors enabling such measurements are now becoming commercially available, but their nontrivial cost of ~ \$5-10M USD routinely limits availability.

3. This work mainly focuses on delithiation process, why did not show a full cycle including lithiation as well, which could better understand the influence of strain variation to particles upon cycling.

Unfortunately, there was not sufficient beamtime to measure the discharge in this experiment. The claims made in this initial work are supported by the data, although more work is needed to understand the complex microstructural evolution in these materials over time.

To understand the influence of cycling on the LMNO crystals, we have added *ex situ* SXDM maps of crystals in the pristine state, and after 60, 90, and 275 cycles, after which the capacity had faded ~20%. 4 to 5 particles were mapped in each state, and we observe significant heterogeneity in their microstructure. The relative misorientation of the domains increases with cycling up to 500%, indicating the slow growth of grain boundaries from a monolithic single crystal. These results are now summarized in Figure 7 in the main text of the revised manuscript, corresponding to maps of each particle shown in Figures S11-14.

In a separate manuscript produced in the last seven months since this study was submitted, we have measured a complete charge/discharge cycle under *operando* conditions. (DOI: 10.26434/chemrxiv-2023-7t1td, also attached). The main objective of that work was to perform simultaneous SXDM and Multi Crystal X-ray Diffraction, which bridges single crystal measurements with powder diffraction on an ensemble of particles.

Figure 4 of the supplementary manuscript displays the tilt map and pole figure of the particle before and after discharge. The shape of the intracrystalline domains in the tilt map are similar, but the color (and pole figure) shows the relative orientation of these domains inside the crystal is completely different. This reorganization of the tilted domains is both distinct from the pattern seen in the strain maps (Fig S3 and S4) during discharge, and also from the pattern seen during the charging cycle, which is rather complicated. In our opinion integrating these results would more than double the length of the present manuscript, which we believe should remain focused on the delithiation mechanism of LMNO during its first phase transition, and the first demonstration of *operando* electrochemical SXDM.

4. Actually, most battery materials belong to polycrystals, e.g. polycrystalline nickel-rich cathodes, there are many grains inside one particle, the anisotropic strains easy resulting in the generation of microcracks, how this in situ SXDM technique to detect and imaging the strains along boundaries for these materials.

The reviewer is completely correct that most high-performance cells use polycrystalline layered materials vs the single crystal morphology investigated here. Liu et al (DOI: 10.1038/s41467-021-26290-z, reference 25 in the main text) recently demonstrated an elegant multimodal study examining the grain boundaries in polycrystalline NiMnCoO₂. We have added additional context in the discussion about the possibilities of SXDM on polycrystalline materials.

There are two major limitations using SXDM for polycrystalline materials:

1) For typical polycrystalline battery materials (~50 nm LFP/LTO, 300nm NMC/NCA) this leads to very few pixels in the image. Ongoing improvements in microscope resolution are now routinely expanding capabilities down to 2-5 nm (see DOI: 10.1364/OE.25.025234), for certain samples. Ultimately, we are limited by beam damage, which increases dramatically at high magnification, comparable to in-situ TEM experiments.

2) SXDM only detects crystallites which happen to be oriented in the Bragg condition, less than 1% of the total. This leads to images where the small primary particles are 'floating in space', with no information on the context of where it resides inside the larger secondary particle. As suggested by the second reviewer, it is possible that simultaneous XRF imaging may allow for better insight here, but is not compatible with conventional electrodes. *Operando* microLaue diffraction is another option, although its much lower detection efficiency produces much larger beam damage, and no group has yet managed to perform such an experiment to the best of our knowledge.

REVIEWERS' COMMENTS

Reviewer #1 (Remarks to the Author):

The authors have addressed all comments of the reviewers in great detail and placed additional information for better understanding inside the manuscript and supporting information. The article reads well and is recommended for publication without further remarks.

Reviewer #2 (Remarks to the Author):

The reviewers addressed all my questions. The current form of the manuscript reads well to the reviewer. The reviewer recommends the manuscript for publication with NC.